# Knowledge, Attitudes and Institutional Readiness towards Social Accountability as Perceived by Medical Students at the University of the West Indies in Trinidad

Bidyadhar Sa [1,*], Christal Patrick [2], Onella Pascall [2], Jalisa Patrick [2], Sade Pierre [2], Diana Pillai [2], Kion Persad [2], Allan Patterson [2], Nicholas Peterson [2] and Reisha Rafeek [3]

[1] Centre for Medical Sciences Education, Faculty of Medical Sciences, The University of the West Indies, St. Augustine 685509, Trinidad and Tobago
[2] Faculty of Medical Sciences, The University of the West Indies, St. Augustine 685509, Trinidad and Tobago
[3] School of Dentistry, Faculty of Medical Sciences, The University of the West Indies, St. Augustine 685509, Trinidad and Tobago
* Correspondence: bidyadhar.sa@sta.uwi.edu

**Abstract:** Background: Social accountability is defined as "the obligation of medical schools to direct their education, research and service activities toward addressing the priority health concerns of the community, region, and/or nation that they have a mandate to serve". It is becoming increasingly critical in evaluating medical school performance and education quality. Medical students must have an appropriate understanding of social accountability. This study explores knowledge, attitudes and institutional readiness as perceived by medical students towards social accountability. Method: An independent online cross-sectional survey was conducted, which included 121 medical students recruited through a convenience sampling technique. The survey instruments were validated through a pilot study and the responses were analyzed using chi-squared ($\chi^2$) tests. Frequencies and percentages were computed. Results: A total of 69% of students understood SA, 61.2% believed they demonstrated SA, and 82.6% believed it has a positive impact on their attitudes and behaviors. About 52.1% believed that their school has a positive impact on the community with a curriculum that includes (52.9%) and reflects the needs of the population that they will serve (49.6%). Conclusion: Based on the findings, a significant number of students have knowledge about social accountability, have a positive attitude towards the concept, and believe that their institution demonstrates readiness.

**Keywords:** social accountability; service learning; medical student; knowledge; attitude



## 1. Introduction

Social accountability (SA) in medical education is becoming increasingly critical in evaluating medical school performance and education quality. SA is defined as "the obligation of medical schools to direct their education, research and service activities toward addressing the priority health concerns of the community, region, and/or nation that they have a mandate to serve" [1]. The concept of social accountability has been further developed through the work of the *2010 Global Consensus for Social Accountability of Medical Schools* document [2]. The document, as presented in Box 1, indicates that the consensus on social accountability embraces a system-wide scope from the identification of health needs to the verification of the effects of medical schools on those needs.

Key parameters (quality, equity, relevance and effectiveness) were provided in a framework laid out by Boelen and Woollard in 2009 [3] for a medical school to be recognized as socially accountable. In 2012, Boelen et al. [4] clarified the indicators to help medical schools to be able to craft benchmarks to assess progress towards social accountability. Many other studies have looked at concepts and frameworks [5–7]. While social accountability is an ideal for medical schools to strive for, it remains a challenge to measure the

societal impact [8]. Medical schools in different developing regions around the world, such as the Eastern Mediterranean Region [9], Sudan [10], Latin America [11] and Korea [12], are striving to enhance their opportunity for accreditation as social accountability is regarded as a benchmark of excellence in medical education and is part of the medical accreditation of schools [10], and the Caribbean region is no different with respect to accreditation.

**Box 1.** Ten (10) SA thematic areas developed by the Global Consensus for Social Accountability of Medical Schools 2010

---

AREA 1: Anticipating Society's Health Needs.
AREA 2: Partnering with the Health System and Other Stakeholders.
AREA 3: Adapting to the Evolving Roles of Doctors and Other Health Professionals.
AREA 4: Fostering Outcome-Based Education.
AREA 5: Creating Responsive and Responsible Governance of the Medical School.
AREA 6: Refining the Scope of Standards for Education, Research and Service Delivery.
AREA 7: Supporting Continuous Quality Improvement in Education, Research and Service Delivery.
AREA 8: Establishing Mandated Mechanisms for Accreditation.
AREA 9: Balancing Global Principles with Context Specificity.
AREA 10: Defining the Role of Society.

---

The Caribbean Accreditation Authority for Education in Medicine and Other Health Professions (CAAM-HP) was established in 2003 under the auspices of the Caribbean Community (CARICOM). CARICOM is a political and economic affiliation of 15 member states, and it includes most of the English-speaking islands and some Central and South American nations. Member countries are Antigua and Barbuda, the Bahamas, Barbados, Belize, Dominica, Grenada, Guyana, Haiti, Jamaica, Montserrat, Saint Kitts and Nevis, Saint Lucia, Saint Vincent and the Grenadines, Suriname, and Trinidad and Tobago. Associate members of CARICOM include Anguilla, Bermuda, the British Virgin Islands, the Cayman Islands, and the Turks and Caicos Islands. The establishment of the CAAM-HP by CARICOM is an integral component of the regional emphasis on ensuring quality medical education [13]. The CAAM-HP was granted World Federation of Medical Education (WFME) Recognition for a ten-year term, from May 2012 to May 2022 and extended to 2023. "Recognition by WFME confers the understanding that the accreditation agency has been assessed and found to be credible in its policies and procedures to assure the quality of medical education in the programs and medical schools that it accredits" [14].

The University of the West Indies (UWI) is a regional university that serves 15 countries with five campuses located in the Anglophone Caribbean. The UWI was founded in 1948 at Mona, Jamaica. It began first as a College of the University of London. In that year, 33 students from nine Caribbean countries were admitted to the founding Faculty of Medicine. In 1961, The UWI became an independent entity and, about that time, it established two other campuses, first in Trinidad and Tobago at St Augustine and later at Cave Hill, Barbados [15]. The University of the West Indies (UWI), Faculty of Medical Sciences (FMS), St Augustine campus (STA), Trinidad and Tobago, began teaching its undergraduate MBBS program in October 1989, and the first batch of medical students graduated in 1994. The faculty has strengthened the human resource capacity in the health sector of most of the contributing territories of the Caribbean region, and the vast majority of the undergraduates have been from Trinidad and Tobago. Thus, the faculty serves a key role in providing health professionals for the region [16]. Furthermore, the mission statement states that the institution's mission is "To advance learning, create knowledge and foster innovation in the Medical Sciences for the positive transformation of the Caribbean and the wider world", which refers to societal transformation. Social accountability is a foundation of the institution and influences why it was set up and how it has been practiced, even though the institution does not explicitly define social accountability. SA has become critical and is demanded in the accreditation standards, and we would like to test our readiness through student surveys so we can better prepare. Besides teaching and research, one of the three dimensions of a university is community outreach and social accountability.

The UWI medical school at STA is committed to that. It could be safely stated that social accountability is becoming increasingly important to medical students and schools around the world, and these institutions and students must be prepared for these changes.

Regarding medical students, social accountability (SA) entails working with the nation to improve the health system while caring for patients and the general population. In a study conducted by McCrea and Murdoch-Eaton, 2014 [17], the results showed that most medical students do not know or understand the concept of social accountability. Although medical school education may have a curriculum geared towards social accountability, students would not be aware of this unless it is directly told to them. Another study highlighted that social accountability is poorly understood and, therefore, there is a need for medical schools to take an external look at the changing needs of society and examine how medical practices can have a positive impact on the health of the population [18]. It is hypothesized that in Trinidad and Tobago, most medical students do not know, understand or recognize the concept of social accountability as they do the term service learning (SL). Service learning is defined as an "educational experience in which students participate in an organized service activity that meets identified community needs" and includes reflection on the service in order to gain an enhanced sense of civic responsibility [19]. Social accountability and service learning complement each other because they both focus on addressing community needs. By encouraging students to serve the underserved communities and address their health concerns, service learning encourages students to understand the idea of social accountability. It has been shown that when students go out to communities for service learning, this is seen as a move towards a social accountable institution [20–22], and, from a teaching perspective, exposing students to service learning in a rural/remote community is a promising model for teaching social accountability [23]. The Liaison Committee on Medical Education of the Association of American Medical Colleges has recognized the potential positive implications of service learning and has added a mandate to their standards that medical schools "provide sufficient opportunities for, encourage, and support medical student participation in service-learning and/or community service activities [24]".

Students value participating in service-learning projects as it is an effective way to integrate into a community, and it is also gratifying to them to participate in serving others [25]. In a narrative review of over 40 studies, it was reported that students indicated that exposure to social accountability activities was a meaningful experience with the potential of positively influencing their future patient care [26]. While medical students may have a curriculum geared towards allowing them to become socially accountable individuals, students may be unaware of this unless it is explicitly stated. Although many might dispute that social accountability has been a part of the medical curriculum for many years, it can be argued that awareness and understanding of this concept are frequently clouded and misplaced. Thus, in addressing this issue, students must consider SA, their knowledge and representations of this concept, and their perceptions of their faculty's involvement in its realization. This study aimed to assess the knowledge, attitudes and institutional readiness perceived by medical students towards social accountability at the University of the West Indies (UWI), St. Augustine Campus (STA).

## 2. Materials and Methods

Ethics Approval: Ethical approval was required to undertake this study and, therefore, all research members completed the Cooperative Institutional Training Initiative (CITI) program. Ethical approval was granted by The Campus Research Ethics Committee and the UWI, Faculty of Medical Sciences, St Augustine Campus, Trinidad (CRECSA.1335/01/2022).

Pilot Study: The survey instruments were validated through a pilot study that included 15 medical students, 2 lecturers, 2 physicians and 1 sociologist. They were approached and recruited via WhatsApp, emails and phone calls as a result of the restrictions and regulation implemented at that time due to the COVID-19 pandemic. Daily classes gave unlimited access to choosing lecturers to participate in the pilot study and recruiting the 2 physicians.

One of the research group members had a former high school teacher who was a sociologist and asked that person if they would like to participate in the pilot study. The 15 medical students were recruited via email addresses from the Dean's office. However, because these students participated in our pilot study, they were not allowed to participate in the official survey administered to our target sample of medical students. This online pilot study was conducted to validate the quality of the final survey by ensuring that it contained questions pertaining to social accountability.

The pilot study's results were successful. The overall design of our survey received a lot of positive feedback and recommended corrections. Some of the feedback included comments such as "Some questions assess attitude but assume that students know what Social Accountability is." Another commenter asked, "Where are the qualitative questions?" and added, "Where there are only Yes/No choices, perhaps consider a "don't know/not sure" option?" Overall, all of the criticism was viewed favorably, and the necessary adjustments were made so that our survey was entirely valid and reliable for our intended audience.

An independent online cross-sectional survey was conducted to assess medical students' knowledge, attitudes, and institutional readiness towards SA. The full study included 121 medical students selected through convenience sampling technique [27]. Medical students in Years 1–5 were recruited via student email addresses provided by the Dean's Office, as well as through coordination with each year's class representatives to aid in the dissemination of the survey directly to their peers. Regarding the nature of the questionnaire, the participants were informed of the nature of the study and ensured anonymity of their responses. The instruments implemented in the questionnaire were a yes-or-no scale (Knowledge about SA); a 5-point scale including the options of strongly agree, agree, neutral, disagree and strongly disagree (Attitudes towards SA) developed by the investigators based on the literature; and a no/somewhat/good/excellent scale (Institutional Readiness) developed by the Training for Health Equity Network (THEnet) and The International Federation of Medical Students' Associations (IFMSA). The Cronbach's alpha for Attitude towards SA was r = 0.73, and for Institutional Readiness, r = 0.86, which is highly acceptable.

The collected data were stored on a password-controlled computer accessible only by the study's researchers. The collected data were input into version 24.0 of the Statistical Package for the Social Sciences (IBM Corporation, Armonk, NY, USA). For the purposes of descriptive statistics, age, age of acceptance into medical school, mean and standard deviation were calculated. To determine the null hypothesis that the distribution was accidental, a Chi-squared test of equality was employed using version 28.0.1.0.

## 3. Results

The response rate is 89%, and the participants' mean age and mean age of acceptance into the medical school are 23.08 ± 3.28 and 20.50 ± 2.86, respectively. As presented in Figure 1, the majority are female (66.9%), live in an urban setting (65.3%) and come from CARICOM countries (96.7%), which reflects the present demographic distribution of the students in this school of medicine.

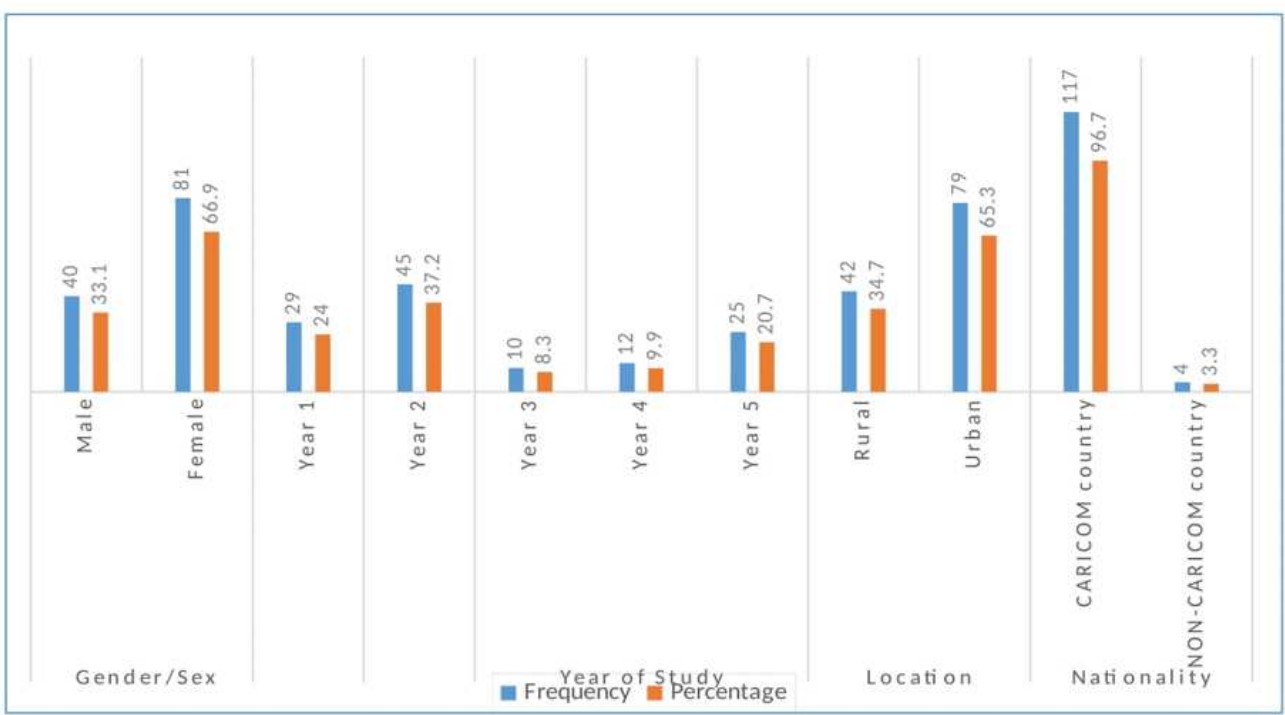

**Figure 1.** Shows the Characteristics of the selected sample Subjects.

### 3.1. Knowledge about the Concept of Social Accountability

More than half of the respondents have heard of the term social accountability or service learning (57%), and nearly three-quarters know that they are not the same thing (74.4%). The majority (86.8%) have the knowledge that SA implies that the school consults with the society to jointly identify priority health issues and expectations; that SA refers to engaging in community-based research projects (72.7%); and that SA ensures that medical institutions produce skillful graduates who are fit for supplying the society's health needs (95%). Table 1 shows the overall sample's responses on knowledge.

As presented in Table 2, a significant number of both male and female students agree that focusing on achieving good grades throughout medical school does not solely relate to being socially accountable (KQ3), $\chi^2 = 9.784$, $p < 0.05$; that SA does imply that the school consults with the society to jointly identify priority health issues and expectations (KQ6), $\chi^2 = 7.222$, $p < 0.01$; and that SA ensures medical institutions produce skillful graduates who are fit for supplying society's health needs (KQ11), $\chi^2 = 7.211$ $p < 0.01$.

A significant number of students across different years of study agree that social accountability encompasses all of the above options (KQ2), $\chi^2 = 24.74$, $p < 0.01$.

**Table 1.** The overall sample's responses on knowledge about the concept of social accountability.

| Sr No. | Item | Yes—*n* (%) | No—*n* (%) | Chi-Square | *p* Value |
|---|---|---|---|---|---|
| KQ1 | Have you ever heard of the concept of 'service learning' or "social accountability" in the UWI or otherwise? | 69 (57) | 52 (43) | 2.388 | 0.122 |
| KQ4 | Do you think service learning and Social Accountability is the same thing? | 31 (25.6) | 90 (74.4) | 28.769 * | 0.001 |
| KQ6 | Social accountability implies that the school consults society to jointly identify priority health issues and expectations. | 105 (86.8) | 16 (13.2) | 65.463 * | 0.001 |
| KQ7 | Social accountability refers to engaging in community-based research projects? | 88 (72.7) | 33 (27.3) | 25.000 * | 0.001 |
| KQ8 | Social Accountability involves working an 8 am–4 pm shift only in hospitals? | 5 (4.1) | 116 (95.9) | 101.826 * | 0.001 |
| KQ10 | Social accountability means prioritizing the health concerns of communities ONLY in a local setting. | 13 (10.7) | 108 (89.3) | 74.587 * | 0.001 |
| KQ11 | Social accountability ensures that medical institutions produce skillful graduates that are fit for supplying society's health needs. | 115 (95.0) | 6 (5.0) | 98.190 * | 0.001 |

| Sr no. | Item | Resolving problems of social health. *n* (%) | Having a mandate to serve the community. *n* (%) | Community engagement through service activities. *n* (%) | All of the above *n* (%) | Chi-Square | *p* value |
|---|---|---|---|---|---|---|---|
| KQ2 | Select the options that describe social accountability? | 21 (17.4) | 20 (16.5) | 4 (3.3) | 76 (62.8) | 98.273 * | 0.001 |

| Sr no. | Item | Being exposed to learning opportunities whereby health services to the underserved are practiced. *n* (%) | Engaging in activities that positively impact the community. *n* (%) | Focusing on achieving good grades throughout medical school. *n* (%) | Participating in research programs. *n* (%) | Chi-Square | *p* value |
|---|---|---|---|---|---|---|---|
| KQ3 | Which answer does NOT solely relate to being socially accountable? | 10 (8.3) | 3 (2.5) | 97 (80.2) | 11 (9.1) | 197.645 * | 0.001 |

| Sr no. | Item | Resolving problems of social health. *n* (%) | Having a mandate to serve the community. *n* (%) | Community engagement through service activities. *n* (%) | All of the above *n* (%) | Chi-Square | *p* value |
|---|---|---|---|---|---|---|---|
| KQ5 | Select the options that describe service learning? | 12 (9.9) | 23 (19.0) | 29 (24.0) | 57 (47.0) | 36.455 * | 0.001 |

| Sr no. | Item | All of the above | Road safety | The way citizens view the health system | Communities | Chi-Square | *p* value |
|---|---|---|---|---|---|---|---|
| KQ9 | Social accountability will positively affect which of the following: | 73 (60.3) | 3 (2.5) | 26 (21.5) | 19 (15.7) | 89.744 * | 0.001 |

For the above responses, the df is 1. * Chi-square values were statistically significant ($p < 0.001$). For the above responses, the df is 3. KQ: Knowledge Question. *Chi-square values were statistically significant ($p < 0.001$).

**Table 2.** Significant background factors versus responses on selected Knowledge Questions about the concept of social accountability.

| | **Knowledge Question KQ3** | | | | |
|---|---|---|---|---|---|
| | Which answer does NOT solely relate to being socially accountable: | | | | |
| | Being exposed to learning opportunities whereby health services to the underserved are practiced *n* (%) | Engaging in activities that positively impact the community *n* (%) | Focusing on achieving good grades throughout medical school *n* (%) | Participating in research programs *n* (%) | Total *n* (%) |
| Sex | | | | | |
| Male | 3 (2.5) | 0 (0.0) | 29 (24.0) | 8 (6.6) | 40 (33.1) |
| Female | 7 (5.8) | 3 (2.5) | 68 (56.2) | 3 (2.5) | 81 (66.9) |
| Total *n* (%) | 10 (8.3) | 3 (2.5) | 97 (80.2) | 11 (9.1) | 121 (100) |
| Chi-squared = 9.784 ***, *** Chi-square values were statistically significant ($p < 0.05$), df = 3 | | | | | |

| | **Knowledge Question KQ6** | | |
|---|---|---|---|
| | Social accountability implies that the school consults society to jointly identify priority health issues and expectations? | | Total |
| Sex | Yes *n* (%) | No *n* (%) | |
| Male | 30 (24.8) | 10 (8.3) | 40 (8.3) |
| Female | 75 (62.0) | 6 (5.0) | 81 (66.9) |
| Total *n* (%) | 105 (86.8) | 16 (13.2) | 121 (100.0) |
| Chi-squared = 7.222 **, ** Chi-square values were statistically significant ($p < 0.01$), df = 1 | | | |

| | **Knowledge Question KQ11** | | |
|---|---|---|---|
| | Social accountability ensures that medical institutions produce skilful graduates that are fit for supplying society's health needs. | | Total *n* (%) |
| Sex | Yes *n* (%) | No *n* (%) | |
| Male | 35 (28.9) | 5 (4.1) | 40 (33.1) |
| Female | 80 (66.1) | 1 (0.8) | 81 (66.9) |
| Total *n* (%) | 115 (95.0) | 6 (5.0) | 121 (100.0) |
| Chi-squared = 7.211 **, ** Chi-square values were statistically significant ($p < 0.01$), df = 1 | | | |

| | **Knowledge Question KQ2** | | | | |
|---|---|---|---|---|---|
| | Select the options that describe social accountability | | | | |
| What is your current Year of Study? | Resolving problems of social health *n* (%) | Having a mandate to serve the community *n* (%) | Community engagement through service activities *n* (%) | All of the above *n* (%) | Total *n* (%) |
| Year 1 | 5 (4.1) | 4 (3.3) | 0 (0.0) | 20 (16.5) | 29 (24.0) |
| Year 2 | 14 (11.6) | 4 (3.3) | 1 (0.8) | 26 (21.5) | 45 (37.2) |
| Year 3 | 0 (0.0) | 4 (3.3) | 0 (0.0) | 6 (5.0) | 10 (8.3) |
| Year 4 | 2 (1.7) | 2 (1.7) | 0 (0.0) | 8 (6.6) | 12 (9.9) |
| Year 5 | 0 (0.0) | 6 (5.0) | 3 (2.5) | 16 (13.2) | 25 (20.7) |
| Total *n* (%) | 21 (17.4) | 20 (16.5) | 4 (3.3) | 76 (62.8) | 121 (100.0) |
| Chi-squared = 24.847 ***, *** Chi-square values were statistically significant ($p < 0.05$), df = 12 | | | | | |

### 3.2. Attitudes towards the Concept of Social Accountability

In terms of attitudes towards SA (Table 3), the majority of the medical students agree or strongly agree that community duties should be mandatory (66.1%); that it should be compulsory for medical students to spend time amongst populations (71.9%); that they have a duty of care to the overall needs of the society (88.4%); that SA helps prepare medical students for a purpose that is fit for society (89.2%); that SA positively impacts the attitude and behavior of medical students (82.6%); and that demonstrating and exercising social accountability is extremely important amongst medical students (85.1%). However,

only 61.2% agree or strongly agree that they themselves have demonstrated SA at that time. The majority strongly disagree or disagree that they would only participate in social accountability or service-learning activities in highly served populations (79.3%) or that, as a medical student, they have a duty of care to the sick only (85.9%).

**Table 3.** The overall sample's attitudes towards the concept of social accountability.

| Sr No. | Item | SDA *n* (%) | D *n* (%) | N *n* (%) | A *n* (%) | SA *n* (%) | Chi-Square | *p* Value |
|--------|------|------|------|------|------|------|------|------|
| AQ1 | Community duties should be mandatory for medical students | 6 (5.0) | 10 (8.3) | 25 (20.7) | 60 (49.6) | 20 (16.5) | 65.736 * | 0.001 |
| AQ2 | Medical students who lack social accountability are not suited to carry out their purpose/future profession | 5 (4.1) | 20 (16.5) | 38 (34.1) | 42 (34.7) | 16 (13.2) | 39.702 * | 0.001 |
| AQ3 | Hospital and clinical duty should be given high priority only. | 11 (9.1) | 58 (47.9) | 33 (27.3) | 16 (13.2) | 3 (2.5) | 78.959 | 0.001 |
| AQ4 | It should be compulsory for medical students to spend time amongst populations. | 2 (1.7) | 9 (7.4) | 23 (19.0) | 60 (49.6) | 27 (22.3) | 83.256 * | 0.001 |
| AQ5 | As a medical student I have a responsibility towards the priority health concerns of society. | 2 (1.7) | 4 (3.3) | 5 (4.1) | 64 (52.9) | 46 (38.0) | 137.554 * | 0.001 |
| AQ6 | I rather voluntarily participate in social accountability or service-learning activities in both highly served populations and underserved populations. | 2 (1.7) | 7 (5.8) | 26 (21.5) | 48 (39.7) | 38 (31.4) | 64.000 * | 0.001 |
| AQ7 | I would only participate in social accountability or service-learning activities in underserved populations | 8 (6.6) | 63 (52.1) | 24 (19.8) | 19 (15.7) | 7 (5.8) | 86.397 * | 0.001 |
| AQ8 | I would only participate in social accountability or service-learning activities in highly served population | 27 (22.3) | 69 (57.0) | 21 (17.4) | 4 (3.3) | - | 75.595 * | 0.001 |
| AQ9 | Social accountability can be used to mold medical students into better health care practitioners for clinical training and development of professional behavior | 2 (1.7) | 3 (2.5) | 5 (4.1) | 59 (48.8) | 52 (43.0) | 136.149 * | 0.001 |
| AQ10 | As a medical student I would have a duty of care to the sick and a duty of care to the overall needs of society. | 3 (2.5) | 2 (1.7) | 9 (7.4) | 46 (38.0) | 61 (50.4) | 124.083 * | 0.001 |

**Table 3.** *Cont.*

| Sr No. | Item | SDA<br>*n* (%) | D<br>*n* (%) | N<br>*n* (%) | A<br>*n* (%) | SA<br>*n* (%) | Chi-Square | *p* Value |
|---|---|---|---|---|---|---|---|---|
| AQ11 | As a medical student I have a duty of care to the sick only. | 32 (26.4) | 72 (59.5) | 9 (7.4) | 5 (4.1) | 3 (2.5) | 140.281 * | 0.001 |
| AQ12 | As a medical student I have a duty of care to the overall health needs of society. | 5 (4.1) | 6 (5.0) | 9 (7.4) | 54 (44.6) | 47 (38.8) | 96.645 * | 0.001 |
| AQ13 | Medical students must actively participate in community activities. | 2 (1.7) | 12 (9.9) | 21 (17.4) | 67 (55.4) | 18 (14.9) | 102.430 * | 0.001 |
| AQ14 | Social accountability helps prepare medical students for a purpose that is fit for society. | 2 (1.7) | - | 11 (9.1) | 77 (63.6) | 31 (25.6) | 110.901 * | 0.001 |
| AQ15 | Demonstrating and exercising social accountability is extremely important amongst medical students | 1 (0.8) | 4 (3.3) | 13 (10.7) | 71 (58.7) | 32 (26.4) | 137.306 * | 0.001 |
| AQ16 | I demonstrate social accountability. | - | 4 (3.3) | 43 (35.5) | 63 (52.1) | 11 (9.1) | 75.860 * | 0.001 |
| AQ17 | Social accountability positively impacts the attitude and behavior of medical students | - | 2 (1.7) | 19 (15.7) | 72 (59.5) | 28 (23.1) | 88.355 * | 0.001 |
| AQ18 | Social accountability negatively impacts the attitude and behavior of medical students | 31 (25.6) | 53 (43.8) | 25 (20.7) | 10 (8.3) | 2 (1.7) | 64.909 * | 0.001 |

For the above responses, the df is 4, except AQ8, AQ14, AQ16 and AQ17 for which the df is 3. AQ: Attitudes Question. * Chi-square values were statistically significant ($p < 0.001$).

As presented in Table 4, a significant number of both male and female students agree that social accountability can be used to mold medical students into better health care practitioners for clinical training and development of professional behavior (AQ9), $\chi^2 = 12.692$, $p < 0.05$. Most disagree that medical students have a duty of care to the sick only (AQ11), $\chi^2 = 9.741$, $p < 0.05$, showing that the students believe that social accountability means more than this. Furthermore, a significant number of both rural and urban students disagree that they would only participate in social accountability or service-learning activities in highly served populations (AQ8), $\chi^2 = 8.282$, $p < 0.05$.

**Table 4.** Significant background factors versus responses on selected Attitude Questions towards the concept of social accountability.

| | | | | | | |
|---|---|---|---|---|---|---|
| **Attitude Question AQ9** | | | | | | |
| Social accountability can be used to mould medical students into better health care practitioners for clinical training and development of professional behaviour | | | | | | Total |
| Sex | Strongly Disagree | Disagree | Neutral | Agree | Strongly Agree | |
| Male | 1 (0.8) | 3 (2.5) | 0 (0.0) | 24 (19.8) | 12 (9.9) | 40 (33.1) |
| Female | 1 (0.8) | 0 (0.0) | 5 (4.1) | 35 (28.9) | 40 (33.1) | 81 (66.9) |
| Total *n* (%) | 2 (1.7) | 3 (2.5) | 5 (4.1) | 59 (48.8) | 52 (43.0) | 121 (100.0) |
| | Chi-squared = 12.692 ***, *** Chi-square values were statistically significant (*p* < 0.05). df = 4 | | | | | |
| **Attitude Question AQ11** | | | | | | |
| As a medical student I have a duty of care to the sick only. | | | | | | Total |
| Sex | Strongly Disagree | Disagree | Neutral | Agree | Strongly Agree | |
| Male | 11 (9.1) | 19 (15.7) | 5 (4.1) | 2 (1.7) | 3 (2.5) | 40 (33.1) |
| Female | 21 (17.4) | 53 (43.8) | 4 (3.3) | 3 (2.5) | 0 (0.0) | 81 (66.9) |
| Total *n* (%) | 32 (26.4) | 72 (59.5) | 9 (7.4) | 5 (4.1) | 3 (2.5) | 121 (100.0) |
| | Chi-square = 9.714 ***, *** Chi-square values were statistically significant (*p* < 0.05), df = 4 | | | | | |
| **Attitude Question AQ8** | | | | | | |
| Residential Location | I would only participate in social accountability or service-learning activities in highly served populations | | | | | Total |
| | Strongly Disagree | Disagree | Neutral | Agree | Strongly Agree | |
| Rural | 14 (11.6) | 19 (15.7) | 6 (5.0) | 3 (2.5) | 0 (0.0) | 42 (34.7) |
| Urban | 13 (10.7) | 50 (41.3) | 15 (12.4) | 1 (0.8) | 0 (0.0) | 79 (65.3) |
| Total *n* (%) | 27 (22.3) | 69 (57.0) | 21 (17.4) | 4 (3.3) | 0 (0.0) | 121 (100.0) |
| | Chi-square = 8.282 ***, *** Chi-square values were statistically significant (*p* < 0.05), df = 3 | | | | | |

### 3.3. Perceptions of Institution's Readiness towards the Concept of Social Accountability

With respect to the medicals students' perceptions of their institution's readiness towards the concept of social accountability (Table 5), 49.6% perceived their curriculum to be good/excellent at reflecting the needs of the population they serve, while 34.7% thought it to be somewhat so. About 52.6% perceived that the places/locations they learn at are good/excellent in including the presence of the populations they serve and, regarding the question 'does your school have a positive impact on the community?' 52.1% perceived it to be good/excellent and 32.2% perceived it to be somewhat so.

In response to the question 'Does your institution have a clear social mission (statement) around the communities that they serve?", 23.1% of medical students responded "No", and 25.6% did not perceive that their teachers reflect the socio-demographic characteristics of the reference population.

As presented in Table 6, across the years of study, a significant number of students somewhat agree that their school has community partners and stakeholders that shape their school (IRQ3), $\chi^2$ = 29.701, *p* < 0.01. Most students agree that the places/locations they learn at (in practice) are good at including the populations that they will serve (IRQ5), $\chi^2$ = 23.092, *p* < 0.05. Most students believe that their learning experience provides a somewhat active service to their community (IRQ9), $\chi^2$ = 27.728, *p* < 0.01, and most students believe that their school has excellent–good community-based research (IRQ10), $\chi^2$ = 25.445, *p* < 0.05.

**Table 5.** The overall sample's perceptions of their institution's readiness towards the concept of social accountability.

| Sr no. | Item | No | Somewhat | Good | Excellent | Chi-Square | *p* Value |
|---|---|---|---|---|---|---|---|
| IRQ1 | Does your institution have a clear social mission (statement) around the communities that they serve? | 28 (23.1) | 55 (45.5) | 35 (28.9) | 3 (2.5) | 45.711 * | 0.001 |
| IRQ2 | Does your curriculum reflect the needs of the population you serve? | 19 (15.7) | 42 (34.7) | 49 (40.5) | 11 (9.1) | 32.620 * | 0.001 |
| IRQ3 | Does your school have community partners and stakeholders who shape your school? | 20 (16.5) | 61 (50.4) | 33 (27.3) | 7 (5.8) | 52.851 * | 0.001 |
| IRQ4 | Do you learn about other cultures and other social circumstances in medical context in your curriculum? | 27 (22.3) | 52 (43.0) | 34 (28.1) | 8 (6.6) | 32.818 * | 0.001 |
| IRQ5 | Do the places/locations you learn at in practice include the presence of the populations that you will serve? | 17 (14.0) | 40 (33.1) | 49 (40.5) | 15 (12.4) | 28.256 * | 0.001 |
| IRQ6 | Are you required to do community-based learning (opposed to only elective opportunities)? | 38 (23.1) | 31 (25.6) | 47 (38.8) | 15 (12.4) | 17.149 * | 0.001 |
| IRQ7 | Does your class reflect the socio-demographic characteristics of your reference population? | 18 (14.9) | 54 (44.6) | 36 (29.8) | 13 (10.7) | 34.537 * | 0.001 |
| IRQ8 | Do your teachers reflect the socio-demographic characteristics of your reference population? | 31 (25.6) | 51 (42.1) | 32 (26.4) | 7 (5.8) | 32.223 * | 0.001 |
| IRQ9 | Does your learning experience also provide an active service to your community? | 23 (19.0) | 57 (47.1) | 30 (24.8) | 11 (9.1) | 37.645 * | 0.001 |
| IRQ10 | Does your school have community-based research? | 7 (5.8) | 42 (34.7) | 53 (43.8) | 19 (15.7) | 43.727 * | 0.001 |
| IRQ11 | Does your school encourage you to undertake generalist specialties (e.g., family medicine, general practice)? | 26 (21.5) | 39 (32.2) | 45 (37.2) | 11 (9.1) | 22.570 * | 0.001 |
| IRQ12 | Does your school have a positive impact on the community? | 19 (15.7) | 39 (32.2) | 48 (39.7) | 15 (12.4) | 24.818 * | 0.001 |

For the above responses, the df is 3. IRQ: Institutional Readiness Question. * Chi-square values were statistically significant ($p < 0.001$).

**Table 6.** Significant difference in the perceptions of students in Years 1–5 regarding their institution's readiness towards social accountability, with selected items shown.

| What is Your Current Year of Study? | Does Your School Have Community Partners and Stakeholders Who Shape Your School? | | | | Total |
|---|---|---|---|---|---|
| | No | Somewhat | Good | Excellent | |
| Year 1 | 4 (3.3) | 10 (8.3) | 10 (8.3) | 5 (4.1) | 29 (24.0) |
| Year 2 | 3 (2.5) | 25 (20.7) | 16 (13.2) | 1 (0.8) | 45 (37.2) |
| Year 3 | 0 (0.0) | 8 (6.6) | 2 (1.7) | 0 (0.0) | 10 (8.3) |
| Year 4 | 5 (4.1) | 6 (5.0) | 1 (0.8) | 0 (0.0) | 12 (9.9) |
| Year 5 | 8 (6.6) | 12 (9.9) | 4 (3.3) | 1 (0.8) | 25 (20.7) |
| Total *n* (%) | 20 (16.5) | 61 (50.4) | 33 (27.3) | 7 (5.7) | 121 (100.0) |

Chi-Square = 29.701 **, ** Chi-square values were statistically significant ($p < 0.01$), df = 12

*Header for Table 6, above the data:* **Institutional Readiness Question IRQ3**

**Table 6.** *Cont*.

| **Institutional Readiness Question IRQ5** | | | | | |
|---|---|---|---|---|---|
| What is your current Year of Study? | Do the places/locations you learn at in practice include the presence of the populations that you will serve? | | | | Total |
| | No | Somewhat | Good | Excellent | |
| Year 1 | 6 (5.0) | 8 (6.6) | 12 (9.9) | 3 (2.5) | 29 (24.0) |
| Year 2 | 8 (6.6) | 19 (15.7) | 15 (12.4) | 3 (2.5) | 45 (37.2) |
| Year 3 | 1 (0.8) | 2 (1.7) | 6 (5.0) | 1 (0.8) | 10 (8.3) |
| Year 4 | 1 (0.8) | 2 (1.7) | 9 (7.4) | 0 (0.0) | 12 (9.9) |
| Year 5 | 1 (0.8) | 9 (7.4) | 7 (5.8) | 8 (6.6) | 25 (20.7) |
| Total *n* (%) | 17 (14.0) | 40 (33.1) | 49 (40.5) | 15 (12.4) | 121 (100.0) |
| Chi-Square = 23.092 ***, *** Chi-square values were statistically significant ($p < 0.05$), df = 12 | | | | | |
| **Institutional Readiness Question IRQ9** | | | | | |
| What is your current Year of Study? | Does your learning experience also provide an active service to your community? | | | | Total |
| | No | Somewhat | Good | Excellent | |
| Year 1 | 3 (2.5) | 13 (10.7) | 7 (5.8) | 6 (5.0) | 29 (24.0) |
| Year 2 | 9 (7.4) | 19 (15.7) | 13 (10.7) | 4 (3.3) | 45 (37.2) |
| Year 3 | 0 (0.0) | 4 (3.3) | 6 (5.0) | 0 (0.0) | 10 (8.3) |
| Year 4 | 1 (0.8) | 8 (6.6) | 3 (2.5) | 0 (0.0) | 12 (9.9) |
| Year 5 | 10 (8.3) | 13 (10.7) | 1 (0.8) | 1 (0.8) | 25 (20.7) |
| Total *n* (%) | 23 (19.0) | 57 (47.1) | 30 (24.8) | 11 (9.1) | 121 (100.0) |
| Chi-square = 27.728 **, ** Chi-square values were statistically significant ($p < 0.01$), df = 12 | | | | | |
| **Institutional Readiness Question IRQ10** | | | | | |
| What is your current Year of Study? | Does your school have community-based research? | | | | Total |
| | No | Somewhat | Good | Excellent | |
| Year 1 | 0 (0.0) | 12 (9.9) | 10 (8.3) | 7 (5.8) | 29 (24.0) |
| Year 2 | 3 (2.5) | 9 (7.4) | 25 (20.7) | 8 (6.6) | 45 (37.2) |
| Year 3 | 0 (0.0) | 2 (1.7) | 5 (4.1) | 3 (2.5) | 10 (8.3) |
| Year 4 | 0 (0.0) | 5 (4.1) | 6 (5.0) | 1 (0.8) | 12 (9.9) |
| Year 5 | 4 (3.3) | 14 (11.6) | 7 (5.8) | 0 (0.0) | 25 (20.7) |
| Total *n* (%) | 7 (5.8) | 42 (34.7) | 53 (43.8) | 19 (15.7) | 121 (100.0) |
| Chi-Square = 25.445 ***, *** Chi-square values were statistically significant ($p < 0.05$) df = 12 | | | | | |

Furthermore, a descriptive statistical analysis on institutional readiness reveals that the overall mean score is $16.12 \pm 6.55$, which is in the second band, i.e., 9–17. It means the school has some social accountability strategies and look for ways to advocate to build on these existing strategies.

## 4. Discussion

Social accountability has been highlighted in 2010 in the *Global Consensus for Social Accountability of Medical Schools* publication, which outlines 10 strategic directions [2]. Since then, medical schools across the world have been assessing whether or not medical students are aware of SA and evaluating their own institutional social responsibility and readiness [28]. Canadian medical schools were early to adapt to and assess SA [29], and the Eastern Mediterranean Region countries have also been active in publications from 2012 to 2022 [30]. The Latin American countries have also engaged in developing a Social Accountability Instrument for Latin America (SAIL) in order to validate its use for assessing social accountability [31]. The three different gradients in the social obligation of medical schools range from social responsibility to social responsiveness and then social accountability [4].

Previous studies have revealed that medical students have limited awareness of the concept of social accountability, and whilst many aspects of undergraduate training should contribute to the acquisition of those key characteristics of social accountability, these would appear to be underdeveloped and not recognized by students [17]. Regarding medical students' knowledge of social accountability, a study conducted in Morocco [18] concluded that 33.5% of the students had heard of social accountability and nearly 79% of the respondents believed that students do not play a significant role in society and that they should concentrate on their education. Another study conducted in a medical college in South India found that 61.6% were not aware of their social accountability [32]. These two studies were at variance to the findings in our study, which found that 57% of medical students had heard of social accountability or service learning, while 95% perceived that SA ensures that medical institutions produce skillful graduates who are fit for supplying society's health needs.

In terms of attitudes towards SA, 61.2% of the respondents believed that they demonstrated social accountability and 82.6% stated that it positively affects the attitudes and behaviors of medical students. In addition, 88.4% of the respondents in this study believed that they have a duty of care to the overall needs of society.

Our findings were in agreement with a study conducted in Saudi Arabia which identified that most medical students perceived their institutions to be socially accountable [26]. In another study by Sebbani et al., 2021 [18], students believed that their school employed social accountability practices, which is similar to the views of the medical students in our study, where 52.1% of the respondents believed that their school has a positive impact on the community and 52.9% believed that their institution's curriculum reflects the needs of the population that they will serve. Nearly a quarter of the respondents did not perceive that their institution has a clear social mission (statement) around the communities that they serve, and this area should be further strengthened. In a previous study, final-year medical students in Saudi Arabia were more critical than students in other years about their institution's social accountability [33]. It is believed that engaging students in community-based learning sites representing the actual population ensure the acquisition of well-defined competencies for more efficient health service delivery and encourage medical students to feel their school's impact on the community and, thereby, improve their perceived SA [34]. Medical students in this study have family medicine clerkships in Year 4 and primary care clerkships in Year 5, and are placed within numerous sites around the island within public hospitals and government health centers for clinical training. This enables their interaction with the actual population in the community. The UWI Medical School at STA has taken steps toward social accountability in medical education and in training students in the needs of society; however, further research is needed to assess the societal impact of the medical school as this remains a challenge to measure [8]. A recent survey across 81 medical schools in 14 countries found that while most respondents expressed an institutional commitment to SA, the effects of their outcomes on the community remains unknown and are not evaluated [35].

A limitation of this study is that the generalizability of the findings to other medical schools may not be applicable. Furthermore, self-reported data have their own biases [36,37]. The responses of the students were collected at one point in time only and cannot ensure that the students will go on to become socially responsible practitioners as they may change during their career. Follow-up studies could be embarked upon to explore how the graduates take it forward to their place of practice.

## 5. Conclusions

In this study, the medical students know about social accountability, are enthusiastic about the concept, and believe that it can shape students into becoming better health care practitioners through clinical training at all levels of public health. The students believe their university has a solid foundation in social accountability by including and reflecting the needs of the community positively. As a result, the students believe that the

institution is prepared to facilitate or teach the concept of social accountability. Based on the findings of this research, a significant number of medical students have knowledge on social accountability, they have a positive attitude towards the concept, and they believe that their institution is ready and equipped to take up the issue of social accountability. This may contribute to increasing critical medical school performance and the scholarship necessary to provide holistic care for their community and nation.

**Author Contributions:** Conceptualization, B.S., C.P., O.P., K.P., S.P., D.P., A.P., J.P. and N.P.; methodology, B.S., C.P., O.P., K.P., S.P., D.P., A.P., J.P. and N.P.; software, B.S.; validation, B.S., C.P., O.P., K.P., S.P., D.P., A.P., J.P. and N.P.; formal analysis, B.S., C.P., O.P., K.P., S.P., D.P., A.P., J.P. and N.P; investigation, B.S., C.P., O.P., K.P., S.P., D.P., A.P., J.P. and N.P.; resources, B.S., C.P., O.P., K.P., S.P., D.P., A.P., J.P. and N.P.; data curation, B.S. and C.P.; writing—original draft preparation, B.S., C.P., O.P., K.P., S.P., D.P., A.P., J.P., N.P. and R.R.; writing—review and editing, B.S., C.P., O.P., K.P., S.P., D.P., A.P., J.P., N.P. and R.R.; visualization, B.S., C.P., O.P., K.P., S.P., D.P.,A.P., J.P. and N.P.; supervision, B.S. and C.P.; project administration, B.S., C.P., O.P., K.P., S.P., D.P., A.P., J.P. and N.P.; All authors have read and agreed to the published version of the manuscript.

**Funding:** This research received no external funding.

**Institutional Review Board Statement:** The study was conducted in accordance with the Declaration of Helsinki, and was approved by the Institutional Review Board (or Ethics Committee) of The Campus Research Ethics Committee and the UWI, Faculty of Medical Sciences, St Augustine Campus, Trinidad (CRECSA.1335/01/2022).

**Informed Consent Statement:** Informed consent was obtained from all subjects involved in the study.

**Data Availability Statement:** All data related to this research will be made available upon reasonable request.

**Acknowledgments:** The authors would like to acknowledge the use of the Institutional Readiness Scale developed by the Training for Health Equity Network (THEnet) and The International Federation of Medical Students' Associations (IFMSA). The authors would like thank all MBBS students who spared their valuable time to participate in this study. The authors acknowledge two lecturers, two physicians and one sociologist who provided their expert judgment on the "Knowledge and Attitude" section of the Questionnaire.

**Conflicts of Interest:** The authors declare no conflict of interest.

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
