# Peer review of "Knowledge, Attitudes and Institutional Readiness towards Social Accountability as Perceived by Medical Students at the University of the West Indies in Trinidad"

_ime, doi:10.3390/ime2010002_

Round 1

Reviewer 1 Report

According to KQ4 from the table 1, majority of participants/students perceive that service learning and social accountability (SA) are not the same thing. However, many cited reference in introduction and discussion uses service learning synonymously as social accountability. When I am reading the paper, I understood as service learning was one of the important means/opportunities to be aware of SA and surface its concept. I assume most people reflect their specific community needs of health through service learnings. I would advise to clarify the relationship between service learning and SA as well as how and what way service learning have helped your students to explicitly appreciate the concept of SA, ultimately affecting institutional readiness. 

Are most graduates in your school stay in your lands? Or they go to the US? I am wondering how graduates have been used their understanding of SA to their post-graduate training and beyond? Are they serving communities or working at public health fields? How have they demonstrated their understanding of SA and taking actions in their mother lands of the medical school? I understand the cross-sectional studies have limits to answer such questions. 

Author Response

Dear Reviewer,

Thank for your valuable comments. A response report has been uploaded.

Sincerely

Bidyadhar Sa

Reviewer 2 Report

The topic is interesting. However, the paper is not currently in publishable shape. The following should be addressed:

·        The title should include the location the study focused on.

·        It is not clear why this study is needed. So authors need a stronger justification for the study.

·        What is the availability sampling technique? This has to be cited. Do authors mean convenience sampling?

·        The analysis has been done properly. However, it is not clear why ethnicity has not been taken into account since the School has students from many different countries.

·        Authors should clearly explain what the findings contribute to scholarship.

Author Response

(The authors gave the same response as above.)

Round 2

Reviewer 2 Report

Thank you for considering my feedback.